# Fifteen Shades of Grey: Combined Analysis of Genome-Wide SNP Data in Steppe and Mediterranean Grey Cattle Sheds New Light on the Molecular Basis of Coat Color

**DOI:** 10.3390/genes11080932

**Published:** 2020-08-13

**Authors:** Gabriele Senczuk, Lorenzo Guerra, Salvatore Mastrangelo, Claudia Campobasso, Kaouadji Zoubeyda, Meghelli Imane, Donata Marletta, Szilvia Kusza, Taki Karsli, Semir Bechir Suheil Gaouar, Fabio Pilla, Elena Ciani

**Affiliations:** 1Dipartimento di Agricoltura, Ambiente e Alimenti, University of Molise, 86100 Campobasso, Italy; g.senczuk@unimol.it (G.S.); pilla@unimol.it (F.P.); 2Dipartimento di Bioscienze, Biotecnologie e Biofarmaceutica, University of Bari, 70125 Bari, Italy; lorenzo.guerra1@uniba.it (L.G.); claudia.campobasso@hotmail.it (C.C.); 3Dipartimento di Scienze Agrarie, Alimentari e Forestali, University of Palermo, 90128 Palermo, Italy; salvatore.mastrangelo@unipa.it; 4Department of Biology, University Abou Bekr Bélkaid, Tlemcen 13000, Algeria; s.gaouar@mail.univ-tlemcen.dz (K.Z.); meghellikaouadji@gmail.com (M.I.); suheilgaouar@gmail.com (S.B.S.G.); 5Dipartimento di Agricoltura, Alimentazione e Ambiente, Università di Catania, 95123 Catania, Italy; d.marletta@unict.it; 6Animal Genetics Laboratory, University of Debrecen, 4032 Debrecen, Hungary; kusza@agr.unideb.hu; 7Department of Animal Science, Akdeniz University, 07070 Antalya, Turkey; takikarsli@akdeniz.edu.tr

**Keywords:** pigmentation, coat color, cattle, hair greying, selection signatures, SNPs

## Abstract

Coat color is among the most distinctive phenotypes in cattle. Worldwide, several breeds share peculiar coat color features such as the presence of a fawn pigmentation of the calf at birth, turning over time to grey, and sexual dichromatism. The aim of this study was to search for polymorphisms under differential selection by contrasting grey cattle breeds displaying the above phenotype with non-grey cattle breeds, and to identify the underlying genes. Using medium-density SNP array genotype data, a multi-cohort F_ST_-outlier approach was adopted for a total of 60 pair-wise comparisons of the 15 grey with 4 non-grey cattle breeds (Angus, Limousin, Charolais, and Holstein), with the latter selected as representative of solid and piebald phenotypes, respectively. Overall, more than 50 candidate genes were detected; almost all were either directly or indirectly involved in pigmentation, and some of them were already known for their role in phenotypes related with hair graying in mammals. Notably, 17 relevant genes, including *SDR16C5*, *MOS*, *SDCBP*, and *NSMAF*, were located in a signal on BTA14 convergently observed in all the four considered scenarios. Overall, the key stages of pigmentation (melanocyte development, melanogenesis, and pigment trafficking/transfer) were all represented among the pleiotropic functions of the candidate genes, suggesting the complex nature of the grey phenotype in cattle.

## 1. Introduction

The vast range in animal color diversity and its dominant role in adaptation and selection has long allured striking interest in evolutionary biology. Coloration is strictly related to an individual’s fitness, being fundamental for sexual signaling, camouflage, physiological processes, UV protection, and defense against parasites [1]. Thus far, many vertebrate target alleles have been associated with color, encompassing more than 150 genes [2,3,4]. Many of these genes are related to the distribution and density of melanin pigments such as eumelanin and pheomelanin, while others are associated with regional variation in the spatial distribution of pigmentation during development pathways [5]. For example, either genomic changes altering the interaction between α-melanocyte stimulating hormone (MSH) and the melanocortin-1 receptor (Mc1r) or mutations in genes involved in the downstream enzymatic reactions are well known to induce a switch in the production of eumelanin and pheomelanin [6,7].

In this context, studies on domestic animals have laid the foundation for insightful topics into the comprehension of genetic pathways underlying coat coloration [8,9,10]. Several factors make domestic animals a compelling model to investigate the genetic basis of coloration. Indeed, since animal domestication, selective human pressure has thinly shaped a plethora of specific traits, including coat color and patterning, giving rise to a colorful biodiversity of domestic breeds. It should be mentioned that historical traces of human-mediated selection of specific color variants is very ancient and well documented. Another relevant advantage in studying domestic animals is related to our knowledge about kinships, which is an important requirement when studying inheritance of traits, facilitating targeted experimental strategies. Moreover, traits that would have been under purifying selection and appearing with low frequency as rare or private mutations in wild populations might rather propagate showing increased frequencies with moderate-to-large effects as a consequence of domestication [11,12]. One of these traits is the hair greying that is relatively common in domestic breeds, but almost absent in wildlife. Adult grey pelage, as observed in many dogs, horses, and cattle breeds, is not fixed throughout the lifetime, but greying is mainly age-related. In humans and horses, for example, greying is related to melanocyte loss during aging, and the process rate and appearance can vary widely among individuals [8,13]. In grey horses, a dominant 4.6-kb duplication in intron 6 of STX17, which encodes syntaxin, has been identified as being responsible for greying with age [14,15]. Such a single duplication has also been suggested to have pleiotropic effects on at least four other traits, including the incidence of melanoma, thus confirming a complex inheritance pattern [16].

All extant domestic cattle belong to two major lineages, *Bos taurus taurus* and *Bos taurus indicus*, derived from independent domestications. However, current cattle biodiversity is not only the result of genetic drift and natural and artificial selection, but also of putative introgression and admixture events between the two cattle lineages, which occurred in almost all continents, giving rise to more than 1100 recognized breeds [17,18,19,20,21]. Such a tremendous diversity is emphasized by a vast array of phenotypic traits such as coat color, body size, behavior, and production traits [22]. Concerning coat color, the switch from fawn to grey is rather common on almost all continents in cattle breeds of the so-called Steppe/Podolian trunk, which is considered a group having the many ancestral traits within taurine [23,24] but with increasing evidence of a remnant indicine introgression [20]. The typical diluted grey coat observed in these animals is considered to confer a better thermoregulation ability in comparison to darker coated cattle due to higher light reflectance [25], while the pigmented skin contributes to protection against ultraviolet radiation.

In this study, we aim to extend our knowledge of the molecular basis of the greying process by performing a selection signature analysis using a medium density single nucleotide polymorphism (SNP) panel on fifteen breeds displaying a grey coat and four displaying “non-grey” phenotypes. This framework allowed us to gain information on candidate genes associated with greying in cattle. While molecular mechanisms related to pigmentation processes have been well-studied, knowledge about the inverse process of “de-pigmentation” remains unresolved and restricted to a few model organisms.

## 2. Materials and Methods

### 2.1. Cattle Pigmentation Phenotypes

The cattle breeds considered in this study share some peculiar coat color features such as: (i) a fawn coat pigmentation of the calf at birth with lightly colored inner sides of the legs and belly, aureoles around the eyes, a muzzle ring, black nasal mucosa, and dark tail tip and hooves; (ii) the coat color turning from fawn to a light grey usually within the first eight months of age; (iii) pigmented skin throughout the entire life; and (iv) more or less pronounced sexual dichromatism, with females (and castrated males) usually displaying a lighter grey coat color compared to bulls, the latter possibly displaying, in some breeds, dark areas mainly around the eyes and the neck. Four breeds were selected as representative of different “non-grey” coat color phenotypes: Angus (solid black), Charolais (solid white), Limousin (solid red), and Holstein (black and white piebald).

### 2.2. Genotypic Data

In total, 514 genotyped animals were considered in this study (Table 1), belonging to fifteen grey cattle breeds (Chianina, Corsa, Croatian Podolian, Garfagnina, Gascon, Guelmoise, Hungarian Grey, Italian Podolian, Marchigiana, Maremmana, Piedmontese, Romagnola, Turkish Grey, Tyrolean Grey, and Ukrainian Grey) and four “non-grey” cattle breeds (Angus, Charolais, Limousin, and Holstein). SNP genotypes were generated in previous studies [18,26,27,28,29,30] using the Illumina BovineSNP50 Genotyping BeadChip, with the exception of the Illumina BovineSNP50 data for Turkish Gray and Hungarian Gray animals that were generated specifically for this study. Quality control procedures were performed using PLINK software v. 1.07 [31]. The following quality control criteria were applied: (i) loci with call rate ≤ 90% (command—*geno 0.1*); (ii) minor allele frequency ≤ 0.01 (command*—maf 0.01*), and individuals with genotyping rate ≤ 90% (command*—mind 0.1*) were removed; and (iii) non-autosomal loci were removed.

### 2.3. Detection of F_ST_-Outlier Markers 

The estimation of pair-wise F_ST_ values from multiple loci and comparison of these values with their corresponding neutral expectations is the basis of many tests aimed at identifying selection signatures based on population differentiation. In particular, the Markov chain Monte Carlo (MCMC)-based method implemented in BayeScan software [32] is capable of generating null distributions of F_ST_ that are robust to population history and structure [33]. The F_ST_-outlier approach implemented in BayeScan version 2.1 [32] was adopted here to detect markers putatively under differential selection pressure in “grey” and “non-grey” cattle breeds, respectively. To this aim, we performed pair-wise comparisons, contrasting one of the 15 grey breeds versus one of the four non-grey breeds in each pair (Table 1). For each cohort (“grey” vs. “non-grey” breed pair), loci that displayed a *q* value (*q*-value) < 0.05 were retained as putatively under selection. We recall that the *q* value of a given locus is the minimum false discovery rate, i.e., the proportion of false positives expected among outlier markers, at which this locus may become significant. Next, we identified loci that differed significantly in multiple pair-wise comparisons, arbitrarily fixing a minimum multiple-occurrence threshold of at least 15 (25%) out of all the possible pair-wise contrasts in order to define chromosome regions putatively under differential selection. For these loci, we then considered a ±250 kbp region as also putatively under selection. The choice of this window size was based on the knowledge that the extent of linkage disequilibrium among markers in the cattle genome does not significantly exceed 500 kbp [34] and was corroborated by a common usage in published selection signature studies in cattle [35,36,37].

### 2.4. Gene Content of Regions Identified as under Selection and Network Analysis

Annotated genes within the genomic regions putatively under selection were obtained by querying the Genome Data Viewer for *Bos taurus* (available online: https://www.ncbi.nlm.nih.gov/genome/gdv/browser/genome/?id=GCF_002263795.1). In order to query the most updated assembly, positions of SNPs in putatively selected regions were updated from the UMD_3.1.1 to the ARS-UCD1.2 assembly. To investigate the biological function and the phenotypes that are known to be affected by each annotated gene, we conducted a comprehensive search in the available literature and public databases. Furthermore, by using the NetworkAnalyst web-based platform [38], we performed an enrichment analysis and a gene-regulatory network analysis (transcription factor–gene interactions) on the candidate genes preliminarily identified using the F_ST_-outlier approach. 

## 3. Results

### 3.1. SNP Loci under Differential Selection

After quality control, the final dataset included a total of 23,313 SNP loci and 514 animals. The overall results of the F_ST_-outlier analysis performed contrasting each grey cattle breed with each of the four selected “non-grey” breeds are reported in Appendix A. Appendix A shows the number of significant pair-wise contrasts detected for each SNP, within as well as across the four tested scenarios while Table 2 presents the most striking signals identified based on the adopted minimum threshold of overall multiple-occurrence. The locus Hapmap49624-BTA-47893, located on the bovine chromosome 2 (BTA2), was significant in 100% of the pair-wise contrasts performed using Limousin as a reference breed. The ±250 kbp region upstream and downstream of this SNP included the genes *PMS1*, *ORMDL1*, *OSGEPL1*, *ANKAR*, *ASNSD1*, *SLC40A1*, *LOC100848294*, and *WDR75*. The possible involvement of these loci in pigmentation is described in Appendix A. Similarly, the locus Hapmap53144-ss46525999 on BTA4 was significant in 100% of the pair-wise contrasts performed using Holstein as the reference breed. The ±250 kbp region upstream and downstream of this SNP included the genes *MYO1G*, *LOC112446527*, *PURB*, *MIR4657*, *H2AFV*, *PPIA*, *ZMIZ2*, *LOC112446406*, *OGDH*, *TMED4*, *DDX56*, *NPC1L1*, *NUDCD3*, *LOC104972146*, *CAMK2B*, and *YKT6*.The possible involvement of these loci in pigmentation is described in Appendix A. Nine SNPs, in a region of BTA14 spanning from 22,781,305 to 25,472,332 bp (Table 2), were significant in a number of pair-wise comparisons ranging between 15 and 36, with all of them showing significant results in at least two different reference breeds. Taken together, the ±250 kbp regions upstream and downstream the above mentioned SNPs included the following genes: *XKR4*, *TRNAT-AGU*, *TMEM68*, *TGS1*, *LYN*, *CHCHD7*, *SDR16C5*, *SDR16C6*, *PENK*, *LOC112449660*, *IMPAD1*, *LOC107133116*, *TOX*, and *TRNAC-GCA*. The possible involvement of these loci in pigmentation is described in Appendix A. The locus ARS-BFGL-NGS-11271 on BTA26 was significant in 13 and 2 pair-wise contrasts performed using Holstein and Angus as reference breeds, respectively. The ±250 kbp region upstream and downstream of this SNP included the genes *LDB1*, *PPRC1*, *LOC112444554*, *NOLC1*, *LOC112444524*, *LOC101902227*, *LOC785229*, *ELOVL3*, *PITX3*, *GBF1*, *NFKB2*, *PSD*, *FBXL15*, *CUEDC2*, *LOC112444535*, *MIR146B*, *MFSD13A*, *ACTR1A*, *SUFU*, and *TRIM8*. The possible involvement of these loci in pigmentation is described in Appendix A. Considering the concentration of SNPs converging in the relatively small region (2.69 Mb) of chromosome 14 (Table 2, Appendix A), we also explored the gene content of the whole genomic interval delimited by the SNPs BTB-01532239 and Hapmap27934-BTC-065223. In addition to the previously mentioned genes, considering the whole interval on BTA14, the genes *RPS20*, *LOC112449628*, *LOC112449630*, *MOS*, *PLAG1*, *FAM110B*, *LOC101902490*, *UBXN2B*, *CYP7A1*, *TRNAG-CCC*, *LOC112449629*, *SDCBP*, *LOC112449508*, and *NSMAF* were added. The possible involvement of these loci in pigmentation is described in Appendix A. 

### 3.2. Gene Enrichment and Gene Regulatory Network Analysis

A gene enrichment analysis was performed on the candidate genes detected as under differential selection (see Appendix A). When querying the Gene Ontology database, a highly significant (*p*-value = 0.00618) enrichment was observed for the “Regulation of myeloid cell differentiation” biological process (Appendix A), to which *LYN*, *PURB*, and *LDB1* genes contributed. Many significant (*p*-value ≤ 0.05) biological processes were related with “Central nervous system development” (*LYN*, *LDB1*, *OGDH*, *PITX3*, *SUFU*), “Generation of neurons” and “Neurogenesis” (*LYN*, *LDB1*, *CAMK2B*, *OGDH*, *PITX3*, *PLAG1*, *PSD*, *SDCBP*), “Neuron differentiation” (*LYN*, *LDB1*, *CAMK2B*, *OGDH*, *PITX3*, *PSD*, *SDCBP*), with “Establishment of vesicle localization”, “Vesicle localization” and “Establishment of organelle localization” (*GBF1* and *YKT6*), with “Exocytosis” (*LYN*, *YKT6*, *PPIA*), with “Ras protein signal transduction” (*GBF1*, *PSD*, *SDCBP*), with “Meiotic cell cycle” (*CAMK2B*, *MOS*, *PMS1*), with “Homeostasis of number of cells” (*LYN*, *LDB1*, *SLC40A1*), with “Ribonucleoprotein complex biogenesis” (*TGS1*, *NOLC1*, *DDX56*), “RRNA processing” and “RRNA metabolic process” (*NOLC1*, *DDX56*), with “Hematopoietic or lymphoid organ development” and “Immune system development” (*LYN*, *PURB*, *LDB1*, *SLC40A1*, *NFKB2*), and with “Ceramide metabolic process” (*NSMAF*, *ORMDL1*) and “Lipid metabolic process” (*LYN*, *NSMAF*, *ORMDL1*, *CYP7A1*, *NPC1L1*, *IMPAD1*, *ELOVL3*, *TGS1*, *SDR16C5*). 

When querying the JASPAR TF binding site profile database [39] for a gene regulatory network analysis, 75 significant (p-value 0.05) pathways were detected. Among the top fifty significant pathways, some were more clearly related to pigment biology and its impairment: transcriptional mis-regulation; MAPK signaling pathway; mitophagy; cellular senescence; apoptosis; neurotrophin signaling pathway; estrogen, prolactin and oxytocin signaling pathways; cAMP signaling pathway; TNF, and NF-kappa B and Wnt signaling pathways (Appendix A).

## 4. Discussion

### 4.1. Genes Identified as Differentially Selected in Grey vs. Non-Grey Cattle Breeds Are Mostly Involved in Pigmentation Biology 

Since the onset of domestication, coat color has been selected by humans producing marked phenotypic diversity in domestic animals. The availability of well-defined breeds with different coat colors offers a compelling opportunity to investigate the genetic basis of this trait. Here, we identified putative genomic regions exposed to significant signatures of differential selection between grey and non-grey cattle breeds. The identified signals mostly encompass genes involved in pigmentation (Appendix A). Moreover, pair-wise breed comparisons converged in showing a signal widely shared across the tested scenarios at chromosome 14, where six genes (*FAM110B*, *UBXN2B*, *CYP7A1*, *SDCBP*, *SDR16C5*, and *NSMAF*) showed more evident implication in pigmentation.

### 4.2. The Multi-Cohort F_ST_-Outlier Method Is a Robust Approach for Selection Signature Detection

In this study, we (i) pair-wise contrasted 15 “grey” cattle breeds each versus four “non-grey” cattle breeds; (ii) retained, for each pair-wise contrast, loci with *q*-value < 0.05 assumed to be putatively under selection; and (iii) applied an arbitrary and intentionally loose (to account for genetic heterogeneity of the considered breeds) across-scenario multi-occurrence threshold (25%). We previously demonstrated the robustness of a similar multi-cohort framework in a study for identification of selection signatures in Merino (and Merino-derived) sheep breeds contrasted with breeds that had no known Merino genetic background [40]. In this study, using the above-described approach, the highly supported selection signature at chromosome 14 mentioned above was identified. The robustness of the adopted procedure was suggested not only by the occurrence, in the BTA14 selection signature, of mostly genes playing a role in pigmentation biology, but also by the fact that two of them, *SDCBP* and *NSMAF*, had been previously shown to be knowingly or possibly implicated in hair graying phenotypes, respectively. Indeed, *SDCBP* (syndecan binding protein) has been identified as one of the dysregulated genes in grey compared to black hair follicles in human premature hair graying patients [41]; *NSMAF* shares a functional domain with the *LYST* (lysosomal trafficking regulator) gene, which is mutated in the Chediak–Higashi syndrome (CHS), a rare disorder characterized, among other features, by childhood occurrence of silvery grey colored hair [42,43,44,45,46]. In mice, a mutant *LYST* allele, named “*grey*” because of the grey coat color of affected mice, was described by Runkel et al. [47]. Besides humans and mice, CHS has also been described in rat, mink, cats, and cattle, and, for all the above species, the involvement of the *LYST* gene is supported. The CHS phenotype has been additionally described in killer whale, fox, tiger, and bison, but no molecular data was generated to confirm the underlying causal mutation. CHS is distinguishable from other hypopigmentation with immunodeficiency syndromes as only CHS is characterized by the presence of enlarged lysosomes in various cell types, as well as the presence of giant melanosomes in melanocytes that prevent the even distribution of melanin [48]. Möhlig et al. [49] showed that NSMAF-deficient cells have enlarged lysosomes, such as LYST-deficient cells, thus allowing for the formulation of the hypothesis that NSMAF may be involved in the occurrence of cattle phenotypes characterized by hair hypopigmentation/greying similarly to those described in CHS.

Interestingly, even when looking at selection signatures observed only within the considered scenarios, results still pointed to the presence of genes strongly related with pigmentation and, in some instances, with known or assumed negative regulation of pigmentation and hair graying (*ORMDL1*, *SLC40A1*, *MYO1G*, *OGDH*, *YKT6*, *LDB1*, *PPRC1*, *ELOVL3*, *MFSD13A*). For those genes where direct or reasonably assumable evidence of involvement in pigmentation biology was not found, we generally found a role in development/physiology of cells of neuronal origin. This was, in our opinion, an additional level of confirmation of the robustness of the adopted approach for selection signature identification. Indeed, it is largely known that melanocytes have a neuronal origin, as they can derive from late-migrating neural crest cells or from Schwann cell precursors [50,51].

### 4.3. A Complex Molecular Architecture Underlies Coat Color and Patterning

The main processes driving pigmentation can be ascribed to three principal stages: melanocyte development from neuronal crest cells (differentiation of neural crest cells from the dorsal neural tube epithelium, migration of neural crest cells, homing of melanocyte-lineage stem cells, and differentiation into melanoblasts), pigment synthesis (melanogenesis), and pigment transfer from melanocytes to keratinocytes [5,52]. Mutations, either structural or regulatory, occurring in genes related to these processes may alter pigmentation phenotypes. Based on the known functions for the candidate genes located on the converging BTA14 selection signature, *SDCBP*, *FAM110B*, *SDR16C5*, *PENK*, *IMPAD1*, *TOX*, *PLAG1*, and *NSMAF* seemed to be more related with early developmental events; *SDCBP* and *CYP7A1* with melanocyte differentiation and melanogenesis; and *SDCBP*, *UBXN2B*, and *NSMAF* with melanin transfer. As for the candidate genes detected in BTA2, BTA4, and BTA26, *ZMIZ2*, *DDX56*, *CAMK2B*, *LYN*, *SDR16C5*, *PENK*, *IMPAD1*, *TOX*, *LDB1*, *PITX3*, *PSD*, *FBXL15*, *SUFU*, and *TRIM8* seemed to be more related with early developmental events, while the remaining genes seemed more related with melanocyte differentiation and function. The genes playing a role in migration of neural crest cells and cell fate determination may be speculated to affect the pigmentation pattern (lighter ventral coat color, eyes aureoles, and muzzle ring) observed in the cattle breeds under study. Genes able to regulate cholesterol/steroid homeostasis (*PPIA*, *ZMIZ2*, *DDX56*, *NPC1L1*, *CUEDC2*, *LYN*, *PENK*, *PLAG1*, *FAM110B*, *CYP7A1*, *NSMAF*) and genes responsive to these molecules (*FAM110B*) may contribute to determining the sexual dimorphism of the considered breeds. However, genes involved in cholesterol and sphingolipid synthesis and metabolism (*ORMDL1*, *ELOVL3*, *NSMAF*, *CYP7A1*) may also impact cell activities, e.g., through lipid rafts (plasma membrane microdomains enriched in cholesterol and sphingolipids that are involved in the lateral compartmentalization of molecules at the cell surface and participates in internalization of ligands and receptors via raft-dependent endocytosis) [53] or through phosphatidylserine-mediated cell cycle signaling, specifically in relation to apoptosis [54].

Besides genes responsive to endogenous stimuli, including oxidative stress (*OGDH*, *PENK*) and nucleotide stress (*DDX56*), we also identified genes possibly responsive to external stimuli, notably ultraviolet radiation and DNA damage (*PMS1*, *OSGEPL1*, *WDR75*, *DDX56*, *YKT6*, *CUEDC2*, *TRIM8*, *LYN*, *PENK*, *IMPAD1*, *MOS*), consistently with the known protective role of melanin produced by melanocytes against UV-induced cell damage. Additional functional categories largely represented among the over fifty candidate genes detected in this study were (i) protein/vesicle trafficking and/or cytoskeletal modifications (*ORMDL1*, *WDR75*, *MYO1G*, *PURB*, *TMED4*, *NPC1L1*, *CAMK2B*, *YKT6*, *GBF1*, *ACTR1A*, *TRIM8*, *LYN*, *MOS*, *FAM110B*, *SDCBP*, *NSMAF*); (ii) apoptosis and senescence-related genes (*ORMDL1*, *OSGEPL1*, *ASNSD1*, *WDR75*, *PPIA*, *OGDH*, *DDX56*, *SUFU*, *ELOVL3*, *XKR4*, *LYN*, *PENK*, *RPS20*, *NSMAF*); (iii) autophagy (*ORMDL1*, *ASNSD1*, *WDR75*, *MYO1G*, *OGDH*, *NPC1L1*, *CAMK2B*, *YKT6*, *LYN*, *IMPAD1*); (iv) lysosomal disorders (*ORMDL1*, *ASNSD1*, *OGDH*, *TMED4*, *NPC1L1*, *YKT6*, *NSMAF*); (v) epithelial–mesenchymal transition (*PPIA*, *OGDH*, *YKT6*, *ELOVL3*, *LYN*, *FAM110B*); (vi) RNA function and protein folding/assembly (*ORMDL1*, *ANKAR*, *PPIA*, *DDX56*, *TGS1*, *RPS20*); (vii) glutamate metabolism and signaling (*ASNSD1*, *OGDH*, *CAMK2B*, *LYN*, *PENK*, *PLAG1*); (viii) mitochondrial biogenesis/function (*OSGEPL1*, *OGDH*, *PPRC1*, *CHCHD7*, *IMPAD1*); (ix) metabolic reprogramming and plasticity (*ASNSD1*, *WDR75*, *PURB*, *OGDH*, *ELOVL3*); (x) ER stress and unfolded protein response (*ORMDL1*, *OSGEPL1*, *ASNSD1*, *RPS20*); (xi) ciliogenesis/ciliary defects (*ANKAR*, *NUDCD3*, *YKT6*, *SUFU*); (xii) proteasomal degradation (*ORMDL1*, *FBXL15*, *FAM110B*); and (xiii) iron metabolism (*OSGEPL1*, *SLC40A1*, *IMPAD1*).

### 4.4. Hair Greying: A Possible Zebuine Heritage in Taurine Cattle Breeds?

Preliminary to this study, we performed a thorough analysis of photographic material available online concerning coat color, coat color changes with age, and coat color patterns of worldwide cattle breeds (for an overview of the consulted online material, please see Appendix A). Based on our results, a gray hair phenotype in adult cattle that are fawn at birth is typically observed in Asiatic zebuine cattle breeds (e.g., Bhagnari, Dajal, Guzerat, Hariana, Hissar, Kankrej, and Tharparkar), in Latin American imported breeds of zebuine origin (e.g., Nellore), in Asiatic grey Steppe cattle (e.g., Hungarian Grey, Ukrainian Grey, and Turkish Grey), in European breeds of the Podolian group (e.g., Podolica, Marchigiana, Romagnola, Chianina, Maremmana, and Croatian Podolian along the two shores of the Adriatic sea), other local breeds from Mediterranean countries (e.g., Corsa and Gascon from France, Piedmontese and Garfagnina from Italy, and Guelmoise from Algeria), and in the phylogenetically close Alpine cattle breeds, Alpine Grey and Tyrolean Grey. Such a large geographic distribution may be consistent with known ancient migration and admixture events during dispersal of cattle out of the domestication center [17,18,20,55,56,57,58,59,60,61] as well as more recent historical migrations between Asia and Europe and in the Mediterranean [27,62], although admixture phenomena among wild taurine and zebuine progenitors may not be ruled out. Interestingly, we found that several of the candidate genes on BTA14 (*XKR4*, *TMEM68*, *TGS1*, *LYN*, *CHCHD7*, *SDR16C5*, *PENK*, *TOX*, *RPS20*, *PLAG1*) had been previously reported to harbor signals of selection/association in taurine/zebuine composite populations or zebuine breeds [63,64,65,66,67,68,69,70,71,72,73,74,75,76,77,78]. This evidence possibly supports the speculation that pigment-related gene variants and phenotypes in grey taurine cattle may represent a heritage of zebuine origin, despite the fact that, for *PLAG1*, an allele with major effects on body size, weight, and reproduction has been shown to be a >1000 year-old-derived allele that increased rapidly in frequency in Northwestern European *B. taurus* between the 16th and 18th centuries, and that was hence introgressed, towards the 19th and 20th centuries, into non-European *B. taurus* and *Bos indicus* breeds likely to increase the stature of modern cattle [79,80].

### 4.5. Gene Enrichment and Regulatory Network Analysis Underpin Pigmentation-Related Processes 

The many biological processes related with the development and function of neuronal cells detected via gene enrichment analysis were consistent with the neuronal origin of melanocytes (see Section 4.2), while biological processes related with vesicle trafficking where consistent with the specialized function of melanocytes that are regarded as lysosomal-related organelles of endosomal origin [81]. The gene regulatory network analysis highlighted signaling pathways, such as TNF, NF-kappa B, and Wnt signaling pathways, which are well known to drive early developmental events (migration and differentiation of melanocyte precursors), and which may play a role in pattern formation during neural crest cell migration. The analysis also highlighted signaling pathways, such as MAPK, that are known to be involved in melanogenesis. Estrogen, prolactin, and oxytocin signaling pathways may point to an endocrine/paracrine control of melanogenesis, consistently with many literature findings, and may explain the sexual dimorphism observed in the cattle breeds considered in this study. Finally, some functional categories pointed toward cellular stress conditions (transcriptional mis-regulation, mitophagy, cellular senescence and apoptosis) that may underly the de-pigmentation phenotype typical of grey cattle breeds.

### 4.6. The Grey Phenotype in Cattle May Share Similar Genetic Features with the Grey Phenotype in Horses and Human Syndromic Hypopigmentation

While many vertebrate genes have been identified to be associated with coat color, the processes leading to greying in vertebrates have been thus far less investigated. Premature hair greying phenotypes have been observed in the human species, and in some cases this trait may be accompanied by a group of associated signs in a syndromic form. This is, for example, the case in Chediak–Higashi syndrome (mutation at the *LYST* gene) and its analog in several animal species; the Griscelli syndrome (dysregulation of the myosin Va, RAB27a, and SLAC2a complex); and the Hermansky–Pudlak syndrome (mutations in several different genes implicated in lysosome-related organelle biogenesis) and its analog in mice. In all the above-mentioned pathological conditions, impaired biogenesis and/or transport of lysosome-related organelles including melanosomes represent the most diagnostic finding [81]. In the equine species, Rosengren Pielberg et al. [14] provided evidence in support of a cis-acting regulatory mutation as responsible for premature hair graying in eight horse breeds (Arabian, Connemara, Icelandic, Lipizzaner, New Forest pony, Shetland pony, Thoroughbred, and Welsh). The mutation was a 4.6-kb duplication in intron 6 of *STX17* (syntaxin-17). Syntaxins contain SNARE domains that mediate vesicle fusion and are involved in intracellular vesicle transport. Notwithstanding, the authors excluded as unlikely the possibility that the mutation in the *STX17* could influence pigmentation by altering melanosome production or transport, based on the observation that hair and skin pigmentation in Gray horses is perfectly normal at birth and dark skin pigmentation is maintained throughout life (similar to the cattle breeds considered in this study). They hence proposed that the mutation may cause the gray phenotype by promoting melanocyte proliferation, possibly via the over-expression of *STX17* and/or its neighboring gene, *NR4A3*, a nuclear hormone receptor implicated in several biological processes, including cell cycle regulation and apoptosis, also observed as a markedly highly expressed gene in melanomas, a common occurrence in Gray horse breeds (but not, at our knowledge, in grey cattle breeds). Indeed, since hair-follicle melanocytes, unlike dermal melanocytes, are terminally differentiated and undergo apoptosis when pigment synthesis of the hair is complete, new melanocytes are recruited from a pool of stem cells to support the pigmentation of new growing hair. Induction of proliferation in dermal melanocytes would thus predispose, in the authors’ view, to melanoma development, while hyperproliferation of hair-follicle melanocytes would cause premature depletion of stem cells and consequent hair greying. We underline here that the hair greying phenotype in horses has some remarkable parallelisms with the phenotype of the grey cattle breeds under study. In both species, hair and skin are pigmented at birth, and while hair greying develops at a young age, dark skin pigmentation is maintained throughout life. Moreover, many candidate genes identified in this study are knowingly or reasonably assumed to be related with the most diagnostic findings in horse and in syndromic human hair greying, i.e., impaired melanosome biogenesis and/or trafficking/transfer (*ANKAR*, *ASNSD1*, *SLC40A1*, *WDR75*, *MYO1G*, *PURB*, *TMED4*, *DDX56*, *NPC1L1*, *CAMK2B*, *LYN*, *ELOVL3*, *PITX3*, *GBF1*, *CUEDC2*, *MFSD13A*, *ACTR1A*, *UBXN2B*, *SDCBP*, *NSMAF*). In addition, the *YKT6* candidate gene on BTA4 is, like *STX17*, a soluble NSF attachment protein receptor (SNARE) protein. 

The results observed in this study did not allow us to unambiguously propend toward the proposed model implying the impaired biogenesis/transport of lysosome-related organelles (human greying syndromes) or the one that hypothesizes hyperproliferation and consequent depletion of hair stem cells (horse hair greying). However, we would like to comment here on the arguments provided by Rosengren Pielberg et al. [14] for excluding that the mutation in the *STX17* gene could influence pigmentation by altering melanosome metabolism and fate based on the observation that hair and skin pigmentation in gray horses is perfectly normal at birth and dark skin pigmentation is maintained throughout life (like in the cattle breeds considered in this study). Indeed, long-lived skin cells, such as neurons and melanocytes, may more strictly depend on autophagy for cellular homeostasis and normal execution of their functions during aging, while rapidly renewing epidermal epithelium may better tolerate suppression of autophagy [82]. This could explain why, in our cattle model, depigmentation is observed in hairs but not in skin, possibly implying altered autophagic patterns in grey cattle hairs. Transition from fawn to depigmented hairs in calves belonging to the grey cattle breeds is generally completed at 3–4 months of age, concomitantly with the first shedding of hair after birth. Hence, the phaeomelanic pigment of calves may well have originated during fetal life under the effect of “environmental” triggers that may cease to be available after birth. Partial and localized re-pigmentation in adult males may be explained by (i) migration of a distinct pool of progenitor cells from epidermal deposits, under the hypothesis of graying due to hyperproliferation and consequent depletion of hair stem cells, or by (ii) mutated “environmental” conditions, possibly related with steroid hormones, to which hair follicles are exposed in mature compared to young males, under the hypothesis of graying due to altered post-natal melanosome metabolism and fate. The latter scenario would also be consistent with the knowledge that hair follicles in different regions of the body respond differently to different androgens [83].

## 5. Conclusions

In conclusion, the results of this study contribute to our knowledge on the molecular mechanisms underlying the hair greying phenotype observed in various cattle breeds worldwide by identifying genome regions under differential selection and candidate genes with congruent functions. Many of the candidate genes detected in differentially selected regions had pleiotropic effects, and altogether they were implicated in all of the three principal stages in pigmentation, i.e., melanocyte development, melanogenesis, and pigment trafficking/transfer, highlighting the complex nature of the grey phenotype in cattle. We cannot exclude that the putative selection signatures detected in this study may have captured general differences in the overall architecture and functioning of skin and skin appendages between grey cattle breeds and the reference breeds in addition to more specific differences in coat patterning and pigmentation, which was our specific goal. Our findings provide useful insights from a less conventional animal model for better understanding of the genetic architecture of hair greying in humans. Moreover, under a climate change scenario, deciphering complex phenotypes such as coat color, which may impact other relevant physiological traits such as UV- and thermo-tolerance, could contribute to improved resilience of cattle breeding under harsh extensive and semi-extensive conditions.

## Figures and Tables

**Table 1 genes-11-00932-t001:** Outline of the experimental design. N, number of genotyped animals.

Test Breed (GREY)	*N*	Reference Breed (non-GREY) *N* = 24	Reference Breed (non-GREY) *N* = 35	Reference Breed (non-GREY) *N* = 20	Reference Breed (non-GREY) *N* = 33
Chianina	23	Holstein	Limousin	Angus	Charolais
Corsa	32	Holstein	Limousin	Angus	Charolais
Croatian Podolian	24	Holstein	Limousin	Angus	Charolais
Garfagnina	23	Holstein	Limousin	Angus	Charolais
Gascon	20	Holstein	Limousin	Angus	Charolais
Guelmoise	24	Holstein	Limousin	Angus	Charolais
Hungarian Grey	24	Holstein	Limousin	Angus	Charolais
Italian Podolian	24	Holstein	Limousin	Angus	Charolais
Marchigiana	22	Holstein	Limousin	Angus	Charolais
Maremmana	24	Holstein	Limousin	Angus	Charolais
Piedmontese	20	Holstein	Limousin	Angus	Charolais
Romagnola	21	Holstein	Limousin	Angus	Charolais
Turkish Grey	23	Holstein	Limousin	Angus	Charolais
Tyrolean Grey	50	Holstein	Limousin	Angus	Charolais
Ukrainian Grey	48	Holstein	Limousin	Angus	Charolais

**Table 2 genes-11-00932-t002:** Results of the F_ST_-outlier analysis performed contrasting 15 “grey” cattle breeds with each of the four selected “non-grey” cattle breeds (Angus, Charolais, Holstein, and Limousin). Only loci detected as significant (*q*-value < 0.05) in at least 15 pair-wise contrasts out of 60 are presented.

CHR	SNP ID	No. OF SIGNIFICANT CONTRASTS							
		ANGUS	CHAROLAIS	HOLSTEIN	LIMOUSIN	OVERALL	POSITION (ARS-UCD1.2)	CONSIDERED INTERVAL	GENES
2	Hapmap49624-BTA-47893	0	0	0	15	15	6760630	6510630-7010630	PMS1, ORMDL1, OSGEPL1, ANKAR, ASNSD1, SLC40A1, LOC100848294, WDR75
4	Hapmap53144-ss46525999	0	0	15	0	15	76874783	76624783-77124783	MYO1G, LOC112446527, PURB, MIR4657, H2AFV, PPIA, ZMIZ2, LOC112446406, OGDH, TMED4, DDX56, NPC1L1, NUDCD3, LOC104972146, CAMK2B, YKT6
14	BTB-01532239	2	10	9	0	21	22781305	22531305-23031305	XKR4, TRNAT-AGU
14	BTB-01530788	3	9	10	9	31	22867321	22617321-23117321	XKR4, TRNAT-AGU, TMEM68, TGS1
14	BTB-00557532	6	10	9	11	36	22986080	22736080-23236080	XKR4, TRNAT-AGU, TMEM68, TGS1, LYN
14	Hapmap46986-BTA-34282	2	3	2	9	16	23630896	23380896-23880896	CHCHD7, SDR16C5, SDR16C6, PENK, LOC112449660, IMPAD1
14	Hapmap46735-BTA-86653	2	12	0	13	27	23725488	23475488-23975488	SDR16C6, PENK, LOC112449660, IMPAD1
14	ARS-BFGL-NGS-36089	1	5	0	10	16	24019648	23769648-24269648	LOC112449660, IMPAD1
14	Hapmap30932-BTC-011225	0	11	0	8	19	25082860	24832860-25332860	LOC107133116, TOX, TRNAC-GCA
14	BTB-01280026	4	9	0	4	17	25354206	25104206-25604206	TOX, TRNAC-GCA
14	Hapmap27934-BTC-065223	0	10	0	5	15	25472332	25222332-25722332	TOX
26	ARS-BFGL-NGS-11271	2	0	13	0	15	23039524	22789524-23289524	LDB1, PPRC1, LOC112444554, NOLC1, LOC112444524, LOC101902227, LOC785229, ELOVL3, PITX3, GBF1, NFKB2, PSD, FBXL15, CUEDC2, LOC112444535, MIR146B, MFSD13A, ACTR1A, SUFU, TRIM8

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
