# Peer review of "Fifteen Shades of Grey: Combined Analysis of Genome-Wide SNP Data in Steppe and Mediterranean Grey Cattle Sheds New Light on the Molecular Basis of Coat Color"

_genes, 2020, doi:10.3390/genes11080932_

Round 1

Reviewer 1 Report

Fifteen shades of grey. Combined analysis of genome-wide SNP data in Steppe and Mediterranean grey cattle shed new light on the molecular bases of coat color

By Seczuk et al,

Elena Ciani, Corresponding author

Comments to the authors—Change the title.  While this title is a cute take off on a a presently popular book, it may not stand the test of time and promises more than the paper delivers.  This work is a comparison of Grey breeds to several continental European breeds, but not to each other.  Just delete the first part.

While the importance of this work is lessened by the fact that the grey breeds are not the most populous ones world wide, understanding coat color for these breeds and of coat color in cattle continues to be of scientific merit.  The importance of the grey breeds in their regions should not be understated, and the curious changes in coat color in many of these breeds is of interest and could help to explain the greying process in humans and other species.  I’m delighted to see a manuscript of work on the grey cattle breeds.  While I am not an expert on GWAS, I interpret the materials and methods to be a sound application in this initial work, and believe that the authors are careful not to interpret results beyond what is actually shown from the SNPs with significant results.

In specific comments, I will offer a few suggestions of editorial importance and also about clarity of presentation of the work.

Line 18-20.     Prefer a bit more clarity.  As written it sounds like the focus is on what causes the changing of color from fawn to grey, rather than simply the breed differences that encompass this phenomena within the grey breeds. 

Line 23-25      “…60 pair-wise comparisons of the 15 grey with 4 non-grey cattle breeds…”

Line 36-38      This would be true in natural selection, but I would argue it is not necessarily so in artificial selection.  Here it may be mainly the preference of the breeder.

Line 38            Is this the wrong reference?  This reference is about diagnosing Chronic Fatigue Syndrome.  A quick read of the article reveals no mention of coloration related to fitness. 

Line 41            “others” not “other”

Line 53            “inheritance of traits” instead of “traits inheritance”.

Line 54            How about “Moreover, traits that would have been under purifying selection and appearing with low frequency as rare or private  mutations in wild populations, might rather propagate showing increased frequencies with moderate-to-large effects as a consequence of domestication, [11, 12].”

Line 65-67      Unclear, please reword.

Line 71            “on almost all continents”

Line 85            Not sure what “…still result scarce” means. 

Line 89            “The grey cattle breeds considered here, share…”

Line 89 to 95   Editor, please determine if a black and white picture of an example breed would be beneficial as a figure.

Line 99            Please indicate in table 1 how many samples for each breed and how the representative SNPs for each breed were determined.  Or mentions that individuals within breeds were identical for SNPs at the loci identified.  Similarly, how were individuals of the 4 comparison breeds selected….surely there were many (millions?) genotyped individuals to select for those comparisons?

Line 104          Was 50K SNP also by Illumina?

Line 105 to 109           The mention of “commands” only makes sense after PLINK software is identified.  Please change to order of these two sentences.

Table 1            Why are the Grey breeds numbered.  I did not see those number references used anywhere else.  Same for reference breed A, B, C, or D.  Why not drop the numbers and report the number of genotyped individuals of each breed instead?

P5, Line 2        Briefly describe Fst-outlier approach, why was it selected?

P5, Line 4-5    “…contrasting one of the 15 grey breeds versus one of the four non-grey breeds in each pair (Table 1).”

P5, Line 7        Delete “We”

P5, Line 9        “Next, we identified loci that differed significantly in multiple pair-waise comparisons, …

P5, Line 10      “…at least six of 15 pair-wise contrasts.  The procedure was then repeated for each of the…”

P5, Line 10      Though you say it was arbitrary, why was 6 used instead of a different arbitrary choice?  Similarly what is the reason for considering +/- 250kb as putativelu under selection?

P5, Line 22      “preliminarily” not “preliminary”….use is as an adverb.

P5, line 32-48  For Charolais, BTA14 is mentioned.  It looks like this has loci for the other breeds too, since it is mentioned later, should this be pointed out for the other breeds as well in this paragraph. 

Table 2            In the title, in several places there is mention of grey cattle or non-grey cattle.  Please change to reflect “cattle breeds” instead of “cattle”.  Same in the continuation of the table.

P9, line 3         “…allowed detection of a single…”

P9, Line 14      Someting is missing in this sentence.  Perhaps “…downstream from the locuson BTA14…”  Same in the caption for Figure 1.

P9, Line 31      Please provide citation or attribution for JASAS TF.

P9, Line 40      Delete “early”

P9, L41            Isn’t it the diversity of well define breeds with different coat colrs that offers the compelling opportunity?  It seems like a step is missing between the first and second sentence.

P10, Line 69    “were” not “where”.

P10, Line 80-82          Reword sentence.

P10, Line 84    “known” instead of renowned”

Reviewer 2 Report

The authors analyzed a wide range of cattle breed SNP data from previously published work to identify polymorphisms under differential selection and the underlying candidate genes that contribute to "greying" in coat pigmentation. Their analysis revealed several key genes that have been implicated in different stages of pigmentation and the results provide a useful guide for future studies aimed at elucidating the molecular mechanisms of greying pigmentation further. I have made recommendations for improvement below.

The title is great but there are a couple of typos: “shed” should be “sheds” since the subject “analysis” is singular. Also, “bases” should be “basis”

L35 – add “an” before “individual’s”

L38 – change to “So far, a great number of vertebrate target alleles have been associated with color, ….”

L51 – change to “human-mediated”

L52 – I’m not sure what authors mean by “testified”, perhaps “highly tested”

L61 – add comma after “process”

L62 – add comma after “STX17”

L63 – change to “has been identified as being responsible for greying with age”

L66 – it’s not clear what “while calf displaying a fawn coloration” means, the wording is confusing here

L71 – add comma after “continent”, also make continent plural

L84 – change sentence to “While molecular mechanisms related to pigmentation processes have been well-studied, knowledge about the inverse process of “de-pigmentation” remains unresolved and restricted to a few model organisms.”

L89 – change to “The cattle breeds considered in this study share some peculiar coat color…..”

L88 – It would be ideal to have this information in figure form so that readers can simply see the range of color pigmentation phenotypes.

L99 – What kind of samples?

P5L9 – change “that resulted to be” to “that were”

Table 2 legend – add comma after “performed”

P9L3 – change “allowed to detect” to “allowed us to detect”

Figure 1 legend – add “to” after “downstream”

P9L40 – delete “early”

P1L43 – add “a” before “significant”

P11L106 – since data are not shown, the authors should provide a bit more detail about how they analyzed photographic material and an estimate of how many photos were analyzed. Furthermore, this is information would be better suited in a figure showing geographic range of these coat color phenotypes and cattle breeds.

P12L145 – change “analogous” to “analogue”

P12L176 – simplify and refine this sentence

P22L181 – change to “…..the results of this study have contributed to our knowledge on the molecular mechanisms…..”
